# Current Trends and Future Perspective of Mesenchymal Stem Cells and Exosomes in Corneal Diseases

**DOI:** 10.3390/ijms20122853

**Published:** 2019-06-12

**Authors:** Hassan Mansoor, Hon Shing Ong, Andri K. Riau, Tisha P. Stanzel, Jodhbir S. Mehta, Gary Hin-Fai Yam

**Affiliations:** 1Tissue Engineering and Stem Cell group, Singapore Eye Research Institute (SERI), Singapore 169856, Singapore; hassan-mansoor@hotmail.com (H.M.); honshing@gmail.com (H.S.O.); andri.k.riau@gmail.com (A.K.R.); tishap@gmail.com (T.P.S.); 2Al Shifa Trust Eye Hospital, Jhelum Road, Rawalpindi, Pakistan; 3Singapore National Eye Centre (SNEC), Singapore 168751, Singapore; 4Eye-ACP, Duke-NUS Graduate Medical School, Singapore 169857, Singapore; 5School of Material Science and Engineering, Nanyang Technological University, Singapore 639977, Singapore

**Keywords:** mesenchymal stem cells, exosomes, inflammation, angiogenesis, immunomodulation, corneal regeneration

## Abstract

The corneal functions (transparency, refractivity and mechanical strength) deteriorate in many corneal diseases but can be restored after corneal transplantation (penetrating and lamellar keratoplasties). However, the global shortage of transplantable donor corneas remains significant and patients are subject to life-long risk of immune response and graft rejection. Various studies have shown the differentiation of multipotent mesenchymal stem cells (MSCs) into various corneal cell types. With the unique properties of immunomodulation, anti-angiogenesis and anti-inflammation, they offer the advantages in corneal reconstruction. These effects are widely mediated by MSC differentiation and paracrine signaling via exosomes. Besides the cell-free nature of exosomes in circumventing the problems of cell-fate control and tumorigenesis, the vesicle content can be genetically modified for optimal therapeutic affinity. The pharmacology and toxicology, xeno-free processing with sustained delivery, scale-up production in compliant to Good Manufacturing Practice regulations, and cost-effectiveness are the current foci of research. Routes of administration via injection, topical and/or engineered bioscaffolds are also explored for its applicability in treating corneal diseases.

## 1. Introduction

The human adult cornea is about 550 μm thick, comprising of five layers. The outermost layer is the corneal epithelium, followed by the Bowman’s membrane, corneal stroma, Descemet’s membrane and the innermost corneal endothelium (Figure 1) [1]. The cornea serves three functions: (1) as a mechanical and chemical barrier protecting inner ocular tissue, (2) as a transparent medium to allow light transmission and (3) light refraction (it provides about 70% of the eye’s refractive power) [2]. Light passes through the cornea and reaches the retina for transduction into neural impulses. Hence the clarity of cornea enables uninterrupted passage of light to the retina.

Loss of vision is a global burden. The number of visually impaired people of all ages is estimated to be 285 million worldwide, with 39 million blind (Global Data on Visual Impairment 2010, World Health Organization) [3]. These patients lose their independence and usually have a poor quality of life. Corneal diseases are a leading cause of visual loss, affecting more than 10 million people. This can be caused by several clinical conditions, including traumatic injury, chemical burns (acid and alkali injury), infections, iatrogenic causes, i.e., limbal stem cell deficiency, age-related degeneration, and corneal dystrophies (Figure 1). All of these conditions can cause defective changes to the cellular and structural components of the cornea [4]. The formation of corneal scars, haze and opacities, as well as corneal edema compromises corneal functions, causing visual deterioration. However, the majority of corneal blindness is preventable, if treated in a timely way. Many patients in under-developed and developing countries have poor access to healthcare and these diseases are often left untreated. The current treatment option of corneal blindness is corneal transplantation, to replace the damaged cornea with a healthy donor cornea (Figure 1). Despite the significant advances in corneal surgery over the past decade, there are issues related to the availability of donor tissue, limited allograft survival, long-term use of immunosuppressants and the need for surgical expertise [5,6]. Many patients do not have access to corneal transplantation due to high surgical and rehabilitation costs. These represent significant financial and logistic burdens, particularly in view of our aging population. It has been estimated that the direct annual health cost due to corneal blindness is more than US$11,000 per person in 2010 in developed countries (data from Canadian Blood Service 2010 Cost Benefit Analysis: Corneal Transplantation; https://blood.ca/sites/default/files/otdt-indx-final-c2a.pdf). Unfortunately, no cost estimate can be made for the developing countries, but the socio-economic burden is expected to be higher. 

Corneal transplantation is the most frequent type of transplantation worldwide and about 180,000 corneal transplants are performed annually [7]. Although the total number of donated eye globes/corneas has been increasing in recent years (there was a rise of 5.2% in 2013 compared to 2012, Eye Bank Association of America), the demand always outstrips the availability of transplantable donor tissue [6]. The global population is expected to rise by 113% in 2030 (and 122% by 2050) and life expectancy will increase at 0.07% annual rate (data from Department of Economic and Social Affairs, UN; https://www.who.int/blindness/data_maps/VIFACTSHEETGLODAT2010full.pdf). As the population lives longer, the demand for corneal transplants will undoubtedly increase, particularly if there is no disruptive treatment technology. This problem will be further exacerbated by the increased global prevalence of diabetes and systemic diseases, which can contribute to increased graft rejection and failure. Hence, alternative solutions, such as regenerative cell-based therapy, should be explored [8]. 

The cornea is an ideal organ for regenerative cell therapy, due to its immune-privilege and avascular nature [9]. The transplanted cells are not as likely to be rejected as in other tissues or organs. Mesenchymal stem cells (MSCs) with regenerative and differentiation capabilities have received much attention among ophthalmologists and visual scientists as an alternative modality in the management of corneal diseases. The paracrine effect of MSCs, mediated by exosomes, has also been suggested for their therapeutic effect. The cell-free nature of exosomes has gained particular interest with respect to its safety.

## 2. Mesenchymal Stem Cells (MSCs)

MSCs are a population of proliferative and multipotent stem cells present in various tissues throughout development. Human fetal MSCs have been found in different fetal tissues, including first-trimester blood and bone marrow [10], or from extraembryonic tissues (placenta, umbilical cord and amniotic fluid) [11]. In adult tissues, MSCs have been isolated from bone marrow, peripheral blood, adipose tissue, dermis, synovium, periosteum, cartilage, skeletal muscle, fallopian tubes, menstrual blood, gingiva and dental tissue, as well as in the eye (such as corneal stroma and trabecular meshwork) [8,12,13]. In general, fetal MSCs contain more primitive phenotypes than those from adult tissues, such as longer and more active telomeres and greater propagation capacity [14]. However, they often require vigorous ex vivo expansion in order to achieve sufficient numbers for therapeutic use and this can lead to replicative senescence and functional decline [15]. Hence, in vitro protocols have been developed to derive MSCs from human pluripotent stem cells, including embryonic stem cells (ESCs) and induced pluripotent stem cells (iPSCs) [16,17]. A summary of MSC sources is shown in Figure 2. According to the criteria proposed by The International Society for Cellular Therapy (ISCT), MSCs can be enriched via their plastic adherence, cell surface expression of CD73 (5’-nucleotidase), CD90 (Thy1) and CD105 (endoglin); and negative detection of CD34, CD45, HLA-DR, CD14 and CD11b (integrin αM chain) expression [18]. MSCs also exhibit differentiation potential into various types of mesenchymal lineages, such as osteoblasts, adipocytes and chondrocytes, both in vitro under defined conditions and in vivo [19]. They possess the ability to proliferate and migrate to the injury sites, and promote wound healing by secreting anti-inflammatory and growth factors [20]. They also interact with innate and acquired immune cells and modulate immune response via paracrine action [21,22].

However, differences have been identified among MSCs from various tissues. In terms of phenotypic markers, bone marrow and adipose MSCs were found to express CD13, 73, 90, 105 and STRO-1, but had different expression of CD34, 49d, 54 and 106 [23,24]. Besides the different tissue origin, the phenotypic variation could also be attributed to the different isolation methods and propagation media and conditions. MSCs obtained from birth-related tissues, including placenta, amnion, umbilical cord and cord blood have greater proliferative and engraftment capacity as well as differentiation potential than MSCs from adult tissues, such as bone marrow [25,26]. Within the umbilical cord tissue, MSCs isolated from whole umbilical cord, from Wharton’s jelly or from cord blood revealed different proteomic profiles, and the umbilical cord MSCs displayed better differentiation for musculoskeletal tissue engineering [27,28]. In addition, MSCs from umbilical cord exhibited higher proliferation capacity than from bone marrow [29]. Adult adipose MSCs also displayed variations with regards to different origins [30]. MSC from subcutaneous fat tissue proliferated faster than those from the omental region [31]. Early senescence of bone marrow MSCs was also detected while adipose MSCs displayed the associated signs at later passages [32]. 

## 3. MSC Mobilization, Migration and Homing in Corneal Changes

Stem cell mobilization, migration and colonization can be induced by injury and inflammation [20,33]. Upon corneal injury, such as trauma and infection, the endogenous bone marrow MSCs are triggered by specific chemo-attractants to mobilize into the peripheral blood. These circulating MSCs migrate to the injury site in the cornea and engraft to promote wound healing [34]. Ye et al. reported the migration and engraftment of intravenously administered bone marrow MSCs in the injured corneal tissue of a murine alkali-burn model [35]. Among chemokines, SDF-1 and substance P have been shown to regulate MSC mobilization and recruitment to the cornea [34]. Moreover, selectin and integrin-mediated leukocyte-like cell adhesion, transmigration and passive entrapment, are potent mechanisms through which MSCs home to tissues [36]. However, the efficiency of MSC homing and engraftment is generally low, owing to the first-pass retention in the lung, liver, kidneys and spleen after systemic administration [33,37]. Much effort has been made to improve MSC migration to target ocular tissues. Administering MSCs through sub-conjunctival injection and co-transplantation on amnion have been shown to improve the local concentrations of MSCs in injured corneal tissue [38,39,40,41]. This potentiates the MSC effect and reduces the application dosage. However, MSC migration and homing seem not necessary for the treatment to take effect. Roddy et al. demonstrated that systemically administered MSCs released TSG-6 (tumor necrosis factor-stimulated gene/protein 6) to reduce corneal inflammatory damage without MSC engraftment [42]. Similarly, the distal effects of MSC have been shown to decrease post-myocardial infarction inflammation and improve cardiac function, through TSG-6 secretion from MSCs embolized in the lung after intravenous delivery [43].

## 4. MSCs in Corneal Regeneration

The therapeutic effect of MSCs in regenerative corneal therapy can be attributed to the direct cell replacement [36] and also by secreting soluble factors to regulate tissue wound repair, inflammation, angiogenesis and immune response [21,44] (Figure 3, Table 1).

### 4.1. Corneal Epithelial Regeneration

The corneal epithelium covers the outermost part of the cornea and consists of 5–7 layers of stratified squamous non-keratinized cells. A healthy epithelium is maintained by a population of epithelial stem cells harbored in the limbal palisades of Vogt [45,46]. A breach in the integrity of the corneal epithelium (caused by physical abrasion/trauma, infection, limbal stem cell deficiency, etc.) causes persistent epithelial defects, corneal ulcer inflammation, neovascularization and opacities, leading to corneal blindness [45]. 

MSCs have been shown to possess the capacity to transdifferentiate into epithelial cells and lineage derived from the neuroectoderm (including astrocytes and neurons) [21]. Mesenchymal-epithelial transition (MET), as well as its counterpart epithelial-mesenchymal transition, play essential roles in tissue re-modelling, organogenesis and tissue repair in human and animal models [47,48,49]. Various mechanical and environmental cues can initiate and propagate MET, from which the mesenchymal cells acquire epithelial properties, including apical-basal polarity, expression of epithelial genes (E-cadherin—a MET hallmark gene—and cytokeratins) and the formation of adherens and tight junctions [47]. In vitro, rabbit bone marrow MSCs co-cultured with rabbit limbal stem cells and/or conditioned media, attained epithelial-like polygonal and cobblestone morphology, and the resultant cells expressed corneal epithelium-specific cytokeratin 3 (CK3) [50]. Rat bone marrow MSCs in co-culture with rat corneal stromal cells resulted in trans-differentiation into epithelial-like cells with CK12 expression [38]. Moreover, human adipose MSCs grown in conditioned media of corneal epithelial cell culture showed up-regulated CK3 and CK12 expression, and the cells appeared epithelial-like [51]. However, in vivo studies of MET are mostly inconclusive. Transplantation of human bone marrow MSCs tissue-engineered on human amnion onto chemically injured rat corneas did not show any CK3-expressing human cells [52]. The application of rabbit bone marrow MSCs, mixed in fibrin gel to rabbit corneal surface injured by alkali-burn, showed epithelial healing and CK3 expression [50]. However, this result was controversial, since both the transplanted cells and the recipient tissues were of rabbit origin, and no cell labelling was included in the experiment.

The use of small molecule chemicals has gained much attention recently to initiate and alter the cellular changes between mesenchymal and epithelial phenotypes, and to regulate cell fate as well as to facilitate target gene reprogramming [53,54]. We reported a MET protocol antagonizing GSK3 (glycogen synthase kinase 3) and TGFβ (transforming growth factor β) pathways (using a combination of valproic acid, A-83-01, CHIR99021, RepSox, tranylcypromine and all-trans retinoic acid) to generate corneal epithelial progenitors (MET-Epi) from human adipose MSCs [55]. These small molecules are cell permeable, non-immunogenic, dose-modifiable, cost-effective and easy to be standardized. These adipose MSC-derived epithelial progenitors expressed E-cadherin, occludin and cytokeratins with the suppression of N-cadherin, indicating MET progression. In vivo transplantation of these progenitors engineered on fibrin gel to rat corneal surface after alkali injury (mimicking total limbal stem cell deficiency) improved corneal transparency and surface stability with the formation of a multilayered epithelium over the injured corneal surface and elevated expression of human EpCAM (epithelial cell adhesion molecule), CK3 and 12 (manuscript in preparation). The sham group consisted of adipose-derived MSCs engineered on a fibrin gel, did not show human CK3/12 expression and had less reduction of corneal haze. Hence, MSCs could be an alternative tissue-engineered cell source for treating corneal epithelial defects and reconstructing the ocular surface.

Cultivated limbal epithelial transplantation (CLET), on a carrier, such as amniotic membrane or fibrin gel, is a surgical alternative for corneal epithelial failure, due to extensive limbal stem cell deficiency [56,57]. This new procedure using ex vivo propagation of autologous or allogeneic epithelial stem cells from a small limbal biopsy has a theoretical advantage over conventional limbal transplantation in stabilizing the ocular surface [58]. The possibility of MSC therapy to achieve similar therapeutic outcomes as CLET has been demonstrated recently. Calonge et al. compared allogeneic bone marrow MSC transplantation with allogeneic CLET and found that both were safe and restored corneal epithelia from LSCD over a one-year post-treatment follow-up [59]. Central corneal epithelial healing was observed in 71.4% of cases after MSC transplants (n = 17) and in 66.7% of cases after CLET (n = 11). This encouraging result has demonstrated MSC therapy as a potential treatment for corneal epithelial injury. However, further studies are needed to support the clinical efficacy, safety and long-term stability of this treatment.

### 4.2. Corneal Stromal Regeneration

The corneal stroma is the thickest corneal layer (~90% of corneal thickness) and is composed of specialized extracellular matrix (ECM) components and collagen fibrils, organized in the form of flattened lamellae running orthogonally to each other. Corneal stromal keratocytes (CSKs) located between the collagenous lamellae are generally quiescent [60]. The regulated spacing and packaging of collagen fibrils in the corneal stroma, along with the peculiar composition of keratan sulfate proteoglycans (keratocan, lumican, mimecan and decorin), stromal crystallins (aldehyde dehydrogenase 1A1 and 3A1, and transketolase) and ECM proteins (such as collagen I and V), play essential roles in maintaining corneal transparency, biomechanics, and integrity [60]. Corneal insults (trauma/disease) cause the death of CSK at the injured site, resulting in reduced proteoglycan synthesis, increased glycation of collagen molecules as well as the breakdown of collagen fibrils. The surviving CSKs near the injury site are activated to become repair-type stromal fibroblasts to aid stromal wound healing. Some fibroblasts even transform into highly contractile myofibroblasts under the synergistic effect of serum and cytokines (such as TGFβ and platelet-derived growth factor, PDGF), eventually generating corneal haze/opacities and scar formation. The presence of dense opacities and persisting scars can interfere with the passage of light, resulting in visual impairment and even blindness. 

There have been reports on the differentiation of bone marrow and umbilical cord lining MSCs into keratocyte-like cells and these papers have shown the potential of MSCs in restoring corneal stromal clarity [61,62,63]. Lumican knockout (Lum^-/-^) mouse model can mimic congenital corneal diseases, involving stromal opacities, secondary to the distorted arrangement of collagen fibrils [64]. Intrastromal injection of human umbilical cord lining MSCs has been shown to rescue defective collagen phenotype and restored corneal thickness and transparency with the transplanted cells assuming CSK phenotype (expression of keratan sulfate-keratocan and lumican) [61]. In addition, the injected cells suppressed inflammatory reaction, as evidenced by minimal leukocyte and macrophage infiltration, and this could promote cell survival with reduced risk of rejection. Likewise, human MSCs from bone marrow, adipose tissue and limbal stroma showed elevated CSK gene expression at both RNA and protein levels when cultured in keratocyte differentiation condition (supplementation of TGFβ3, bFGF and ascorbic acid), respectively [63,65,66,67]. The transplantation of rabbit adipose MSCs grown on polylactic-co-glycolic acid bioscaffold repaired the mechanically induced corneal stromal defects, without triggering corneal neovascularization and induced the expression of ALDH1A1 and keratocan [68]. 

Corneal stromal stem cells (CSSCs) from limbal stroma display similar properties as MSC with positive expression of Pax6 and MSC markers CD73 and CD90 [69,70]. They have been shown to possess the capability to regenerate transparent stromal tissue, suppress corneal inflammation and reduce scarring by mediating neutrophil infiltration after wounding [69,71,72]. CSSCs can differentiate into CSKs, with the deposition of native stroma-like ECM, when cultured in serum-free condition supplemented with bFGF and TGFβ3, indicating their capacity for stromal regeneration [71,73]. Intrastromal injection of human CSSCs to corneas of lumican-null mouse restored the collagen fibril defects and stromal thickness, leading to a recovery of corneal transparency [69]. Currently, a clinical trial with transplanting cultivated allogenic limbal stromal stem cells, as a treatment for patients with unilateral superficial corneal scars, secondary to bacterial/fungal keratitis or trauma is ongoing in L.V. Prasad Eye Institute, India (National Clinical Trial NCT #03295292). Since cryopreserved CSSCs remain viable and retain various marker expression (including CD73, CD90, CD105, STRO1, and CD166) and potency of differentiation, they serve as an alternative tool for stromal cell replacement in treating corneal opacities [72,74]. 

Dental MSCs have also received much attention in regenerative medicine due to their easy accessibility, high plasticity and minimal ethical issues [75,76]. Among the known dental stem cells, periodontal ligament stem cells (PDLSCs) and dental pulp stem cells (DPSCs) originate from the cranial neural crest and share similar developmental pathway as CSKs [77,78]. Intrastromal transplantation of human DPSCs to mouse corneal stroma adopted CSK phenotype by expressing collagen I and keratocan, and maintained corneal transparency and stromal volume [79]. PDLSCs, on the other hand, express markers of MSC, embryonic and neural stem cells, and they exhibit multilineage potential in differentiating to adipocytes, chondrocytes, osteoblasts and neurons [78,80,81,82]. Our recent study has shown a robust two-step protocol involving spheroid formation and induction by growth factors and cytokines in a stromal niche, to differentiate human PDLSCs towards CSK-like cells with the expression of a broad spectrum of CSK markers [83]. Further work using animal models will delineate the functionality of these cells and assess their translational potential for stromal diseases.

In a pilot clinical trial, stromal cell therapy using autologous adipose MSC was reported in treating patients with advanced keratoconus [84]. This study evaluated a six-month safety outcome of autologous adipose MSC injection with minimal intraoperative or postoperative complications. The manifest refraction and topographic keratometry remained stable and the keratoconic eyes had improved visual function, central corneal thickness and corneal clarity, along with new stromal collagen production (patchy hyper-reflective areas seen on corneal optical coherence tomography). Another clinical trial implanting autologous adipose MSCs with or without sheets of decellularized donor corneal stromal lamina to 11 patients with advanced keratoconus had shown a full recovery of corneal transparency within three months post-surgery [85]. Both studies thus have demonstrated that stromal enhancement by MSC therapy could be effective for the treatment of advanced keratoconic eyes. Further work with larger sample size and longer follow-up will confirm these results. 

### 4.3. Corneal Endothelial Reconstruction

The corneal endothelium consists of a monolayer of hexagonal cells covering the posterior corneal surface [86]. Mature corneal endothelial cells are metabolically active, with continuous ATPase pump activity for the fluid-coupled active transport of ions from the corneal stroma to the aqueous humor. This regulates the stromal hydration level and prevents edema, maintaining corneal deturgescence, necessary for normal vision [87]. These cells are non-mitotic and have limited regenerative capacity, due to the expression of negative cell cycle regulators (such as CIP, INK and p53 protein families), cell contact inhibition and the presence of mitogenic inhibitors (such as TGFβ) in aqueous humor [88]. Hence the progressive loss through aging and traumatic loss from diseases and injury lead to corneal endothelial dysfunction and stromal edema, resulting in visual loss. Corneal transplantation is the main treatment strategy; however, cell injection therapy has been reported recently [89]. These options require fresh human corneas and, with a worldwide shortage of donors, there is a need to find a new functional corneal endothelial cell source for future clinical application. 

MSCs could serve as a potential source to generate corneal endothelial cells for the treatment of corneal endothelial diseases, like Fuchs’ endothelial dystrophy and aphakic/pseudophakic bullous keratopathy. However, the research so far sparingly supports the endothelial differentiation potential of MSCs [90]. Recently, the differentiation of human umbilical cord lining MSCs into corneal endothelial-like cells through specific GSK3β inhibition was reported [91]. Subsequent transplantation to rabbit eyes with bullous keratopathy restored the corneal transparency and thickness, and the new endothelium expressed functional endothelial marker, Na^+^K^+^ATPase. Since there are limited data, more extensive studies are necessary to demonstrate the potential of MSC in corneal endothelial regeneration.

## 5. MSCs in Corneal Inflammation and Angiogenesis 

MSCs are known to possess potent anti-inflammatory and angiogenic-regulatory properties that allow them to have therapeutic potential for corneal diseases. Topical and/or sub-conjunctival administration of bone marrow MSCs has reduced corneal inflammation and angiogenesis after chemical injuries in murine models [35,38,52]. The treatment suppressed the infiltration of inflammatory cells and CD68+ macrophages into corneas, downregulated pro-inflammatory cytokines, including interleukin-2 (IL-2), IL-17, monocyte chemoattractant protein-1 (MCP-1), interferon γ (IFNγ), macrophage inflammatory protein-1α (MIP-1α) and matrix metallopeptidase 2 (MMP2), and pro-angiogenic factors, including vascular endothelial growth factor (VEGF) and bFGF. In addition, MSC treatment up-regulated the anti-inflammatory molecules (IL-6, IL-10, TGFβ1 and TSG-6) and anti-angiogenic mediators (thrombospondin-1 TSP-1 and pentraxin-3) [35,38,42,52]. Therefore, there is a modification of the pro-inflammatory milieu of the injured corneas, and restoration of the environment for corneal epithelial regeneration, which promotes ocular surface wound healing [40].

Angiogenesis is regulated by a myriad of pro-angiogenic (like VEGF, bFGF) and anti-angiogenic factors (including TSP-1, pigment epithelium-derived factor PEDF). Depending on tissue microenvironment, MSCs exert pro- and anti-angiogenic functions. They release angiogenic mediators in a tumorous or ischemic niche leading to the formation of granulation matrix for endothelial cell growth and migration [40,92]. However, MSCs can also upregulate TSP-1 to inhibit angiogenesis by disrupting CD47 and VEGF receptor-2 signaling, and suppressing VEGF–Akt–eNOS pathway [93,94]. In addition, the expression of TSP-1 promotes endothelial cell apoptosis and downregulates MMP2, which is a potent inflammatory chemokine with pro-angiogenic activity [95]. 

Recently, corneal stromal stem cells (CSSCs), from limbal stroma, have been shown to possess modulatory activities for corneal inflammation and scarring [96]. When applied to an acute corneal wound model in mice, they inhibited neutrophil infiltration and downregulated the expression of fibrotic markers, such as tenascin-C, α-smooth muscle actin and SPARC (secreted protein acidic and rich in cysteine), via TSG-6 pathway [72]. Similarly, corneal mesenchymal stromal cells (CMSCs), which are also identified in the limbal stroma, exhibited anti-angiogenic action via PEDF and sFLT-1 (soluble fms-like tyrosine kinase-1) expression and suppressed macrophage infiltration in murine corneas [97]. Intravenous CMSC administration inhibited suture-induced corneal neovascularization, through downregulating VEGF-C, VEGF-D, Tek, mannose receptor C type-1 and 2 inside the stromal matrix, causing delayed recruitment of pro-angiogenic monocytes/macrophages [98].

## 6. MSCs in Corneal Transplantation

Several reports have illustrated the roles of MSC in promoting graft survival [95,99,100]. MSCs, with potent immunomodulatory properties, suppress the maturation and activation of antigen presenting cells (APCs) and dendritic cells, as well as cytotoxicity of natural killer cells [101]. They also induce IL10 secretion from immature APCs, and the conversion into T-cell inhibitors [100,102]. The reduced mature APC population has an additional effect on MSC treatment by enhancing the immune tolerance to allografts [99,100,103]. MSCs also inhibit the proliferation and cytokine secretion of T-cells and maturation of B cells [104]. Together with modulating the regulatory T-cell (Treg) generation, they maintain the graft tolerance and survival of allografts [105,106]. This was demonstrated when MSCs were systemically infused prior to transplantations to prolong the graft survival in mouse models [99,100]. In addition, MSCs possess cell surface glycocalyx, rich in anti-inflammatory molecules, like TSG-6, versican, pentraxin-3 and heavy chain-modified hyaluronan matrix, which modulate the host inflammatory responses [106,107]. 

## 7. Challenges of MSC Therapy

In the 1980s, most early phase clinical trials of MSC therapy have shown the ease to manufacture MSCs from different accessible tissue sources of normal volunteers and to apply these cells for clinical usage. Since then, unregulated industry of stem cell clinics developed, growing from a few businesses in 2008 to nearly 600 in 2016 across the US [108]. Some clinics operated without the US FDA oversight on a technicality and provided misleading information on medical board review and safety precaution. In a recent case report, three patients self-paid the procedures of adipose MSC therapy for age-related macular degeneration in a stem cell clinic, however the consent and procedures were not under the context of any approved clinical trial [109]. After injection, they presented severe visual loss with ocular hypertension, hemorrhagic retinopathy, vitreous hemorrhage, combined traction and rhegmatogenous retinal detachment, or lens dislocation. At one-year post-injection, the retina of one patient became markedly atrophic and had no light perception while the other two recognized only hand motion. These consecutive cases have aroused concerns about the performance of procedures in clinics that charge patients for services and lack clinical and preclinical data to support their practices. Hence, the need of regulatory oversight of these clinics and education of patients is of great importance to protect patients amid the advancing stem cell research and innovation [110,111].

After more than two decades of clinical research using MSCs, the major question remains whether MSCs can fulfill the promises of therapeutic efficacy and life-long safety foreshadowed by preclinical animal studies. Most knowledge of MSC mechanisms of action and safety data are based on preclinical work in animal systems and in vitro studies of human MSCs with reports predominantly providing unambiguous data of the positive effect of MSCs. However, whether MSC effects on animal outcomes are readily translated to equivalency in human clinical trials is debatable. Galipeau et al. have proposed that such dissonance could be due to the discrepancies of immune compatibility, dosing, adaptability and potency of human versus animal MSCs (review in Reference [112]).

The use of varied MSC protocols has also been increasingly concerned over the insufficient reproducibility of experimental findings [113]. Details of different procedures and parameters, including the status of tissues for MSC isolation, sorting, ex vivo expansion, purification, phenotyping, administration of cell products as well as follow-up examinations and tests, need to be fully documented in order to translate the therapeutic potential into a reproducible clinical efficacy and outcomes [114,115].

Another concern is the inherent tendency of MSC to transform into cells of different lineages during ex vivo expansion. Henceforth, it is crucial to control and document the entire MSC expansion procedure to ensure the reproducibility of the process [114]. It is also imperative to establish protocols to monitor the quality of the expanded cells prior to use in therapy, by performing extensive functional characterization, in addition to determining their yield, purity, viability and differentiation [116]. A set of robust potency assays has to be developed for measuring the aforementioned peculiar characteristics of MSCs since this will guarantee the reproducibility of the manufacturing process and improve the predictability of the clinical efficacy of MSC. 

## 8. MSC Potency for Translational Use

To maximize the efficacy of MSCs, a good prediction of their therapeutic abilities will help the identification of the best-qualified cells to be used. The Centre for Biologics Evaluation and Research of US Food and Drug Administration has formulated regulatory policies on safety, purity and potency of cell therapy products (https://www.fda.gov/vaccines-blood-biologics/cellular-gene-therapy-products), however, there has not been any consensus to assess and standardize the potency assessments. Most research groups and clinics measure cell counts, cell viability, colony forming units and differentiation capacity together with immunophenotyping using tissue-specific markers to estimate MSC potency [117,118]. These assays provide information on the presence of healthy MSCs, but not the functional efficacy. Further complementary and reproducible assays tailor-made for a specific application are in need to assess the quality of cells and determine which cultures are the best for the treatment purpose. These include the potential performance of cells to exhibit long-term engraftment, and the ability to generate extracellular ground substances in preclinical models (such as spongy granulation tissue in cutaneous wound repair; collagen and keratan sulfate proteoglycans in corneal repair; vascularized granulation tissue in myocardial infarction treatment) [19,62,119,120]. In addition, MSC secretome in response to the local microenvironment has been actively explored in different biological and clinical contexts (including modulating immune system, inhibiting cell death and fibrosis, stimulating vascularization, promoting tissue remodeling and recruiting resident cells) [121,122]. The biochemical and molecular analyses using MSC-conditioned media or purified MSC-derived extracellular vesicles will distinguish unique signature of cytokines and soluble factors (including proteins and microRNAs), underscoring the functional identification and therapeutic potential of MSCs [123]. Together with the expression array for genes in responder cells, this will capture the effector pathways significant to MSC application. Chinnadural et al. analyzed the secretome changes and transcriptome matrix response of MSCs to peripheral blood mononuclear cells and interferon-γ, and they identified cytokine changes in relation to T-cell regulation [124]. Hence, interrogating the secretome and transcriptomic dynamic response of MSCs to defined microenvironmental or chemical cues could serve to reveal the fitness and potency of MSC products for translational applications. 

## 9. Paracrine Action of MSCs

The therapeutic effect of MSCs in regenerative medicine can be attributed by secreting soluble factors, that regulate tissue wound repair, inflammation, angiogenesis and immune response [21,44]. After intravenous delivery to the body, most MSCs are accumulated within the filtering organs, such as lung, liver and spleen. Nevertheless, many studies have shown that MSCs can regulate tissue repair, even though only a small fraction of engraftment at the site of tissue injury [36]. In one study, subconjunctival MSC injection to alkali-injured corneas promoted corneal wound healing, despite the MSCs remaining in the subconjunctival space [40]. Topical administration of MSCs or conditioned MSC media to a murine corneal epithelial wounding mode, attenuated corneal inflammation, reduced neovascularization and promoted wound healing [125]. With the retention of most MSCs in the corneal stroma rather than epithelium, this indicated that MSCs acted through a paracrine mechanism, rather than direct cell replacement. The soluble factors released from MSCs are mediated through the extracellular vesicles or exosomes [8,44,126,127,128,129,130] (Figure 4).

Exosomes are extracellular vesicles that are produced in the endosomal compartment of most eukaryotic cells [131]. Multivesicular bodies fuse with cell surface plasma membrane and release the intraluminal vesicles (exosomes) to the extracellular region. Exosomes are present in most biological tissues and fluids (including blood, urine, cerebrospinal fluid and sweat) [132,133]. In cell culture, exosomes are released into the conditioned media by cultured cells [134]. Various methods have been established to isolate exosomes, including differential centrifugation, density gradient centrifugation, filtration, size exclusion chromatography, polymer-based precipitation, immunological separation and sieving [135]. The sizes of exosomes are restricted by the multivesicular bodies in the parental cells and are generally considered in nanometer range (30 to several hundred nm in diameter). Within lipid bilayered membrane, the luminal content of exosomes contains proteins, lipids and nucleic acids (DNA, mRNA, miRNAs, long noncoding RNAs). However, the exact composition and content of the exosomal cargo released by different cell types are hard to establish, due to differences in the cellular conditions. Cell homeostasis is the major factor controlling the exosome cargo and content, hence the exosomes isolated from different cell types will display characteristics reflecting the cellular origin [136,137]. The mechanisms of sorting exosomal cargo molecules are poorly understood. Ubiquitination is one of the sorting signals for protein entry into exosomes [138,139]. Ubiquitinated proteins are recognized by receptors, such as ESCRT (Endosomal Sorting Complex Required for Transport) subunits that are involved in the binding and directing cargo towards intracellular endosomal vesicles. Usually, these vesicles contain proteins that regulate its biogenesis, e.g., ESCRT components like tetraspanins (CD63, CD81, CD9) and TSG10, as well as proteins involved in their secretion (like RAB27A, RAB11, ARF6) [140]. From ExoCarta (a web-based resource of exosome bioinformatics; http://exocarta.ludwig.edu.au), the top 10 proteins are heat shock protein 8 (HSPA8), CD63, β-actin (ACTB), glyceraldehyde-3-phosphate dehydrogenase (GAPDH), enolase 1α, cytosolic heat shock protein 90α, CD9, CD81, tyrosine 3-monooxygenase/ tryptophan 5-monooxygenase activation protein, zeta polypeptide (YWHAZ), muscle pyruvate kinase (PKM2) [141,142]. Several functional groups are enriched in exosomes, such as tetraspanins (CD9, CD63, CD81), heat shock proteins (HSC70, HSC90), membrane transporters (GTPase) and lipid-bound proteins. Tetraspanins (CD9, CD63, CD81) are common exosomal markers, which are involved in the production of exosomes. Besides the usage as isolation and enrichment markers, these proteins have been reported in the diagnostics of tumors and infectious diseases. An example is CD63+ exosomes, which are significantly enriched in patients with melanoma and other cancers [143]. Exosomal CD81 has been found to be elevated in the serum of patients with chronic hepatitis C [144]. The exosomal mRNAs, lncRNAs and miRNAs usually reflect the physiological conditions of the parental cells. Exosomal miR-21 and miR-1246 are enriched in the secretion of esophageal cancer cells with metastasis [145,146]. Exosomal miR-9, 107, 124, 128, 134 and 137 in the biological fluid have been associated with Alzheimer’s disease [147]. The exosomes are uptaken by cells through several mechanisms, including (1) ligand–receptor binding with cleavage by extracellular proteases and subsequent release of exosomal contents; (2) direct fusion to the plasma membrane and release of the exosomal cargo into the cytoplasm, and (3) receptor-mediated endocytosis, as well as (4) phagocytosis [148,149,150].

## 10. MSC-derived Exosomes (MSC-Exo)

MSC-Exo encapsulate and transfer biomolecules that modify cell and tissue metabolism, including differentiation, inflammation, angiogenesis, immunosuppression, neurogenesis and synaptogenesis [44,151,152,153]. The cargo proteins also act as specific cell type markers and can regulate recipient cell recruitment, proliferation and differentiation. In vitro, exosomes from human adipose MSCs suppressed the proliferation of macrophages, dendritic cells and natural killer cells, as well as downregulated IFNγ release from effector T-cells [126]. MSC-Exo can be an exciting next-generation approach for treating various pathological conditions. A number of preclinical studies had demonstrated the feasibility and efficacy of MSC-Exo therapy using animal models. Bone marrow MSC-Exo promoted angiogenesis in an infarcted rat myocardium model by stimulating endothelial cell proliferation and suppressed inflammation through inhibiting CD3-stimulated T-cells [122]. The expression of IL10, TGFβ1 and human leukocyte antigen-G in bone marrow-MSC-Exo had anti-inflammatory and immunomodulatory activities against the refractory graft-versus-host disease [127]. Likewise, induced pluripotent stem cell-derived MSC-Exo attenuated limbal ischemia in mice by supporting angiogenesis through upregulation of angiogenesis-related VEGF and angiogenin [154]. The application of umbilical cord-MSC-Exo recovered cisplatin-induced acute kidney injury in rats by inhibiting p38 mitogen-activated protein kinase pathway [155]. Additionally, the same exosome type alleviated carbon-tetrachloride-induced hepatic fibrosis in a mouse model by downregulating Smad2, TGFβ1, collagen I and III [156]. Moreover, bone marrow MSC-Exo salvaged the cognitive impairment in mice after traumatic brain injury by decreasing IL1β levels [157]. It has also been reported that this exosome type restored alveolar-fluid clearance, and improved the hemodynamic parameters of human lungs that were rejected for transplantation [158].

## 11. Application of MSC-Exo on Ocular Tissues

Several studies have evaluated the effect of MSC-Exo on ocular tissues. The periocular injection of human umbilical cord MSC-Exo to a rat experimental autoimmune uveitis (EAU) model decreased the infiltration and migration of leukocytes, macrophages and natural killer cells by downregulating MCP1/CCL21 and MYD88-dependent pathways [159]. These cells expressing Gr-1, CD68, CD161, CD4, IFNγ and IL17, and restored retinal function. Yu et al. also reported that intravitreal injection of exosomes from umbilical or adipose MSC cultures modified the pro-inflammatory milieu in laser-induced retinal injury, by inhibiting MCP1, ICAM-1 (intercellular adhesion molecule-1) and TNFα, hence improving visual function [160]. In a study by Zhang et al., intravitreal injection of human umbilical cord MSC-Exo improved hyperglycemia-induced retinal inflammation in diabetic rats, by transferring miR-126, which suppressed HMGB1 (high-mobility group box 1) signaling [161]. In another rat model of blue-light induced retinal damage, intravitreal umbilical cord MSC-Exo injection showed a dose-dependent suppression of choroidal neovascularization by downregulating VEGFA and inhibiting NFκB pathway, possibly through miR-16 transfer [162,163]. In addition, intravitreal injection of bone marrow MSC-Exo to a rat optic nerve crush model stimulated retinal ganglion cell growth through argonaute-2 signaling, which stabilized miR-16 activity from RNase digestion [164,165]. Since intravenous MSC administration caused similar restoration of retinal functions in EAU and laser-induced retinal injury models [166,167], it is conceivable that the therapeutic effects could be mediated through MSC-Exo action. In a recent clinical trial of five patients with refractory macular holes, the intravitreal injection of human umbilical cord MSC-Exo promoted anatomical and functional recovery; however, one patient experienced an inflammatory reaction [168].

## 12. MSC-Exo Application on Corneal Tissues

Corneal wounds caused by chemical or thermal burns, traumatic injury, and/or immune and hereditary disorders are associated with inflammation, neovascularization, ulceration and scarring. Improper or delayed treatment may lead to blindness. Over the last decade, MSC therapy has been proposed as a possible treatment strategy for anti-inflammation, anti-angiogenesis and immunomodulatory activities. The paracrine action of MSCs has been shown to facilitate tissue wound repair and suppress inflammation and angiogenesis in several tissue models. This cell-free strategy may also exert a significant impact in promoting corneal wound repair, which involves the participation of different factors that modulate inflammation, angiogenesis and tissue regeneration. A few studies have demonstrated the therapeutic functions of soluble factors present in MSC-Exo on corneal wound models in vitro and in vivo. Rabbit corneal stromal cells, when cultured in the presence of rabbit adipose MSC-Exo, showed greater proliferation with less apoptosis, along with the deposition of new ECM proteins (including collagen) [169]. In a murine superficial stromal wound model, topical CSSC-derived exosomes suppressed corneal inflammation and corneal scarring by inhibiting neutrophil infiltration through TSG-6 dependent pathway and downregulated fibrotic markers, including tenascin-C, ACTA2, Col3A1 and SPARC [72]. Moreover, murine corneal epithelial wound healing was promoted by exosomes from human corneal mesenchymal stromal cells [170]. Umbilical cord MSC-Exo carrying β-glucuronidase reduced the accumulated glycosaminoglycans of a mouse mucopolysaccharidosis model, thereby reducing corneal haze [41]. These data have highlighted the potential of the therapeutic use of MSC-Exo in ocular surface diseases and congenital corneal metabolic disorders. 

## 13. The Cargo of MSC-Exo and Potential Mechanisms for Therapeutic Effects

Several proteomic studies have shown that MSC-Exo contain several hundreds to thousands of proteins, depending on the parental MSC type [171,172]. The cell-type-associated abundance and composition of exosomal proteins have been demonstrated. Neprilysin, a β-amyloid-degrading enzyme, was expressed at four-fold higher levels in adipose MSC-Exo than in bone marrow MSC-Exo [173]. The application of exosomes from umbilical cord MSCs and bone marrow MSCs promoted U87MG glioblastoma cell apoptosis, while adipose MSC-Exo supported cell growth [174]. Hence, exosomes from different MSC types are not equivalent, and their activity must be fully identified before considering their therapeutic applications. Indeed, much is still unknown with regard to which components of the exosomal cargo are responsible for the various observed therapeutic effects.

More recently, the importance of miRNAs in MSC-Exo have been described. The exosomal cargo contains a variety of miRNAs, which regulate various events related to angiogenic, inflammatory and neurogenic processes [152,175]. The miRNA profiling work has suggested that several miRNAs in exosomes could serve as a tool to qualify cultured human corneal endothelial cells for cell injection therapy [176]. Reduced miRNA content in exosomes has been described by the knockdown of Alix protein in CSSCs, without any changes in the levels of cellular miRNAs. When these exosomes with reduced miRNAs were topically applied to a murine superficial stromal wound model, they failed to suppress stromal scarring and inflammation, while clear corneas were observed with control exosomes without Alix knockdown [72]; Funderburgh et al. Association for Research in Vision and Ophthalmology (ARVO) 2019 Annual Meeting]. These results support the hypothesis that exosomal-related functions could be mediated through miRNAs. Whether any particular individual or groups of miRNAs are engaged in specific tissue functions need to be explored. A previous study has also shown that MSC-Exo carrying miR-16 downregulated VEGF and suppressed angiogenesis in a breast cancer model [162]. Recently, Fafian-Labora et al. reported the age-related reduced expression of miR-21-5p in bone marrow MSC-Exo (significantly higher in pre-pubertal group than in adults) and this could be associated to age-dependent differences in MSC immune profile through Toll-like receptor 4-mediated signaling [177].

## 14. Storage of Exosomes

There is currently no established protocol for the storage of exosomes. It is imperative to consider the effects of freeze-thaw, storage temperature and buffer on the stability of exosomes in terms of the lipid membrane integrity and therapeutic properties. There has been a mixed outcome regarding the stability of exosomes in sub-zero storage. One study has shown that −20 °C freeze-thaw cycles in PBS did not affect the size of MSC-Exo or impair the exosomal membrane integrity, but there was a significant reduction in vesicle size after two days of storage at 37 °C and after three days at 4 °C [178]. Webber et al. further added that the exosomes, frozen at −20 °C for as long as six months, did not alter the biochemical activity [179]. A study on neutrophil-derived exosomes, however, showed a reduction in vesicle size at −20 °C storage, but not at −80 °C for four weeks [180]. The addition of protease inhibitors before freezing of urinary exosomes at −20 °C did not prevent the loss of exosomal biochemical activity, while freezing at −80 °C resulted in near complete recovery of activity even after seven months of storage [181]. In a more recent study, Bosch et al. demonstrated that the addition of 25mM of trehalose could further improve the exosomal membrane and biochemical function stability [182]. These studies have demonstrated that exosomes can be stored long-term at an extremely low temperature with high recovery without compromising the bioactivity. The possibility to lyophilize exosomes in the presence of trehalose to allow storage stability of the vesicles at room temperature provides ease of handling and transport [183]. As a whole, unlike stem cells, which are sensitive to the storage conditions and can exhibit impaired therapeutic properties as a result of freeze-thaw cycles, improper storage temperature and absence of cryopreservative, exosomes are substantially more amenable to storage and transport, a key consideration from a translational point of view [184].

## 15. Sustained Delivery of Exosomes

The intended biological effects of exosomes can only be produced as a result of the vesicle uptake by target cells via an endocytosis pathway [185]; therefore, the ability to deliver exosomes locally in a sustained release manner is crucial to maximizing their therapeutic application. Studies have shown that direct intravenous, intraperitoneal or subcutaneous injection of exosomes, resulted in rapid clearance from the blood circulation and accumulation in the liver, spleen, lung and gastrointestinal tract [146,186]. Regardless of the delivery route and cell source, the majority of systemically injected exosomes are rapidly taken up by macrophages in the reticuloendothelial system and ejected from the body [187]. Although the fate of exosomes following a subconjunctival injection or a topical application in the anterior portion of the eye has not been explored, we can assume that similar rapid clearance of these nanovesicles would also occur. Topical use of drugs on the ocular surface has always resulted in low drug bioavailability due to the presence of epithelial barrier and rapid tear turnover [188]. To compensate for this, it is often necessary to increase the dosage of drug, which requires further optimization. Alternatively, epithelial debridement to improve drug penetration could be carried out, but this increases the susceptibility of the ocular surface to external microbes.

Another problem that further advocates the need for a sustained delivery system of exosomes is the difficulty in producing the vesicles not only on a large scale but also in high purity and consistent quality. The yield of exosome is typically less than 1 µg of exosomal protein from 1 mL of culture medium, whereas the application dose of exosomes is 10–100 µg of protein [189]. In humans, the effective dose could be in an order of magnitude or more to compensate for the rapid clearance of exosomes from the body. A delivery vehicle, like a hydrogel implant, could be used to load exosomes and prolong the intended therapeutic effect over time (Figure 5). The implant will prevent the encapsulated exosomes from being cleared too rapidly, and also enable the possibility to administer a more localized and concentrated dosage by inserting the implant in the proximity of target sites.

Research into encapsulating exosomes in a delivery vehicle is still in its infancy; hence, we have reviewed materials used to encapsulate exosomes that are derived not only from the MSC culture but also from blood plasma. The same methods or materials to encapsulate exosomes could be applicable regardless of the cell source. Qin et al. was the first to describe the idea of encapsulating exosomes in a hydrogel [190]. They used HyStem®-HP hydrogel, which is a thiol-crosslinked composite containing hyaluronic acid, heparin and gelatin, to load bone marrow MSC-Exo to promote bone regeneration. In addition, sodium alginate hydrogel loaded with exosomes isolated from platelet-rich plasma showed a sustained release of exosomes, resulting in a more efficient skin wound healing in the diabetic rat model, compared to the direct application of exosomes on the wound [191]. In a separate study of rat diabetic skin wound model, chitosan hydrogel loaded with exosomes derived from miR-125-3p-overexpressing synovial MSCs improved the healing response [192]. Patching of exosome-loaded hydrogel over the skin wound revealed faster healing rate, compared to untreated and sham-treated with hydrogel only. For application on the cornea, a biodegradable film/hydrogel loaded with exosomes can be used as a patch over the bare stroma. These biomaterials can be tuned to degrade as the epithelial cells heal over the bare stroma so as not to impede vision for a prolonged period. If intact corneal epithelium has to be preserved from the onset of treatment, the exosomes-loaded film/hydrogel can be implanted intrastromally. 

## 16. Perspective

In different pre-clinical experiments, together with completed and ongoing clinical trials, MSC therapy has emerged as a promising strategy for treating various corneal disorders via tissue repair and soluble factor communications. The risk of tumorigenesis and rejection is relatively low due to the immune-privilege properties of the corneas. However, cell-free therapy using MSC-Exo might constitute an alternative in view from regulatory authorities. Studies using non-ocular models have indicated that exosomes act as an important mediator of cellular information exchange. The cellular uptake of exosome cargoes containing proteins and miRNAs reduces inflammation, regulates angiogenesis and immune-modulation, rescues wound repair, and improves functional recovery. Autologous serum eye drops have been used in the treatment of severe dry eyes, persistent epithelial defects, chemical injuries, and neurotrophic keratopathy [193]. The usefulness of the next-generation biological eye drops that contain MSC-Exo in enhancing corneal repair, and ameliorating corneal inflammation and neovascularization will be instrumental in corneal disease management. MSC therapy could be offered to patients to replace the damaged/diseased corneal cells/tissues, whereas topical MSC-Exo may be applied as an adjunctive therapy for corneal inflammation and/or neovascularization. Besides, the cell-free “off-the-shelf” MSC-Exo eye drops may translate into relatively safer products due to the low risk of toxicity and immunological rejection. Furthermore, MSC therapy can be coupled with topical MSC-Exo to potentiate the biological effects of MSC. 

Being cell-free nanoparticles, MSC-Exo circumvent the problems associated with the transfer of mutated or damaged DNA, immunological rejection, and avoid the first-pass effect (lodging in capillaries of lungs, liver, kidneys) that is associated with systemic MSC application. Additionally, MSC-Exo serve a potential to be used as biological carriers for therapeutic agents. In vitro and in vivo studies have shown that exosomes secreted by GATA4-overexpressing MSC act as a repository of anti-apoptotic microRNAs for cardioprotection in rats [194]. Although MSCs can act as depots for the release of various factors in an extended period, they are prone to change or damage under adverse conditions. This is different for exosomes, of which the cell-free property avoids any chance of cell transformation, and this ensures the consistency of soluble factor types and abundance applied to the target tissues. This offers regulatory advantages over MSCs in clinical use [195]. The major drawback of exosomes is the rapid clearance from the body [196]. It is proposed that macrophages of the mononuclear phagocytic system play an important role in the clearance of intravenously administered exosomes from circulation via the recognition by scavenger receptor [196,197]. In a recent study, PKH26-labelled exosomes obtained from murine melanoma cells were readily taken up by hepatic and splenic macrophages, and endothelial cells of lungs. On the contrary, the clearance of exosomes was delayed in macrophage-depleted mice, thus highlighting the importance of macrophage-mediated exosome uptake [197]. In addition, other factors could influence the in vivo clearance of exosomes, such as exosome membrane proteins (integrins, tetraspanins, etc.), size and content of exosomes, and their surface electric charge [196,197]. Hence, exogenous MSC-Exo delivery will likely require a higher circulatory ‘dose’ for an extended period as compared to cellular MSC therapy or require encapsulation in a biodegradable scaffold, which allows sustained local delivery of exosomes.

Furthermore, establishing standardized/reproducible Good Manufacturing Practice (GMP) and potency assays, determining the effective therapeutic dosing and frequency, evolving pertinent storage practices, and studying the effects of lyophilization, dry storage, freezing and thawing during delivery as well as the potency of exosomes, are essential tasks before clinical trials. It remains to be seen if higher dosage levels of circulating MSC-Exo pose any risk of toxicity in the body. In addition, it is still elusive as to which factors secreted by MSC-Exo are responsible for which particular therapeutic effect, as there is a paucity of literature studying the complete proteomics and genomics of MSC-Exo that are derived from different sources [171,198]. 

## 17. Conclusions

MSCs are multipotent cells, easy to expand in vitro, and have the potential to be differentiated into corneal cell types. They can play important therapeutic roles in corneal reconstruction and restoration of corneal functions through their immunomodulatory, anti-angiogenic and anti-inflammatory properties. MSC-Exo mediate the paracrine action of MSCs and its application in the biomedical field is appealing due to the potential in avoiding the problems of tumorigenesis and cell-fate control. 

## Figures and Tables

**Figure 1 ijms-20-02853-f001:**
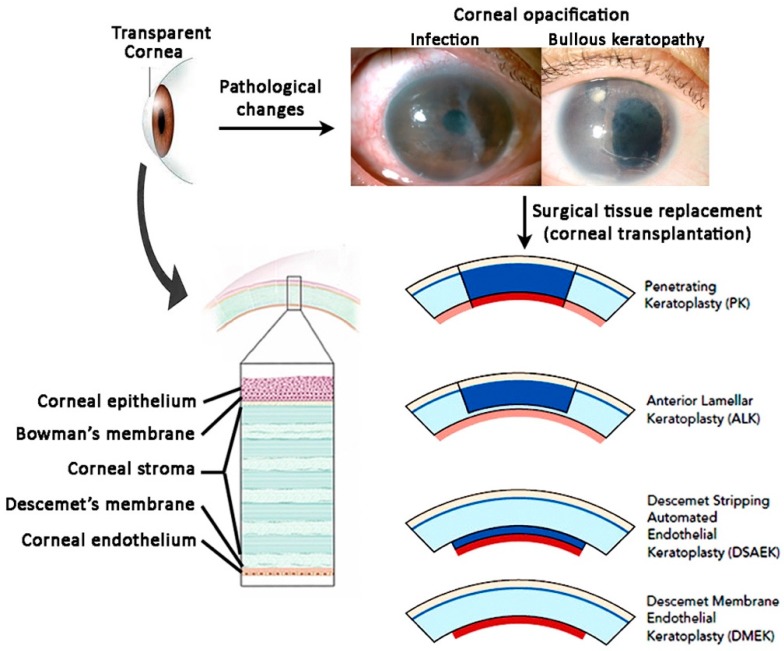
Overview of human cornea and its pathological opacification. Transparent cornea is composed of corneal epithelium, Bowman’s membrane, corneal stroma, Descemet’s membrane and corneal endothelium. Corneal pathologies (e.g., infection, ulcer, injuries) lead to corneal opacification, which is conventionally treated by surgical removal and donor corneal transplantation (penetrating and lamellar keratoplasties).

**Figure 2 ijms-20-02853-f002:**
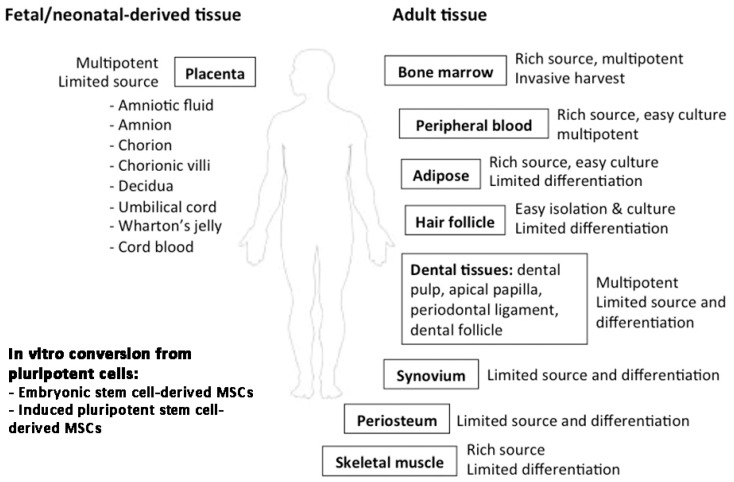
Major MSC sources in human tissues. MSCs can be harvested from (1) fetal/neonatal birth-associated tissues, including placenta (amnion, chorion, decidua), umbilical cord and cord blood; (2) adult tissues, including the rich source of bone marrow, peripheral blood, adipose tissue and limited source from hair follicle, dental tissue, skeletal muscle, etc.; and (3) in vitro conversion from pluripotent cells.

**Figure 3 ijms-20-02853-f003:**
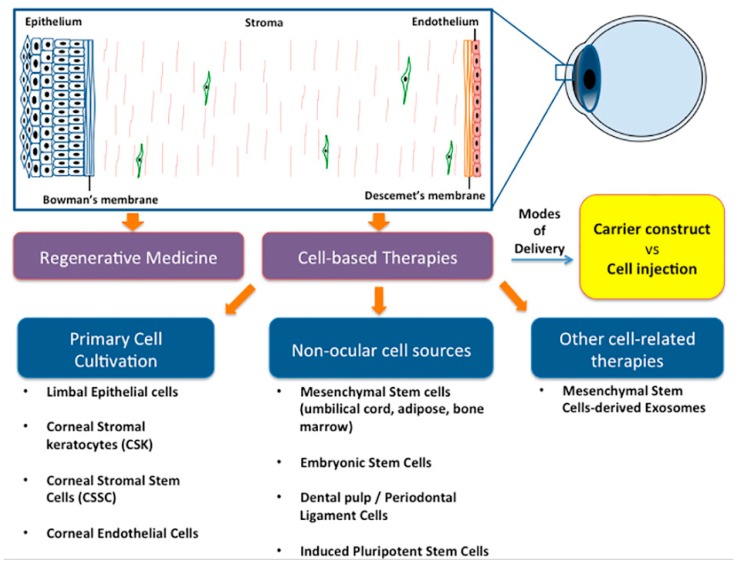
Different approaches of corneal regenerative medicine. Various clinical conditions, including infections, inflammation, traumatic and chemical injuries, age-related degenerations, genetic disorders and corneal dystrophies, can compromise corneal functions by depleting and/or damaging different corneal cells, leading to visual deterioration. Corneal regenerative therapy by primary cell cultivation, transdifferentiation of non-ocular cell sources, and through mesenchymal stem cell-derived exosomes (MSC-Exo) is gaining importance. The cultivated healthy cells can be delivered to patients either by direct cell injection or by seeding them on carrier scaffolds followed by transplantation to restore corneal functions.

**Figure 4 ijms-20-02853-f004:**
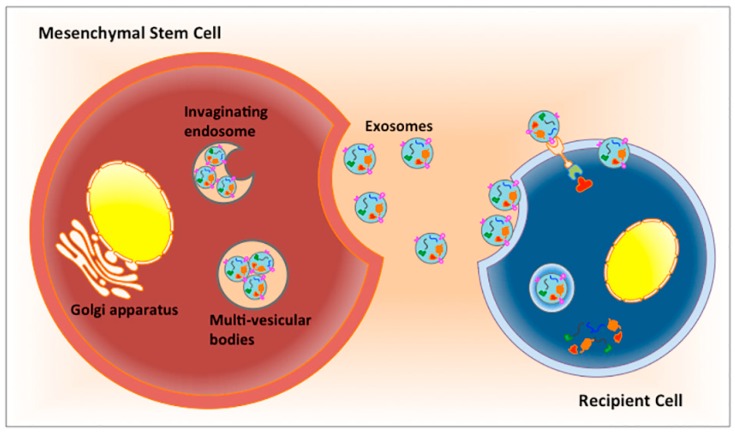
Illustration showing the paracrine signaling mechanism of MSCs. Exosomes are packaged within multivesicular bodies of MSC. The fusion with the plasma membrane results in their release from parental cells via exocytosis. The cargo content of exosomes is diverse in nature and contains lipids, proteins, as well as nucleic acids. Exosomes mediate their effects in the recipient cells by direct fusion and releasing their content into the cytosol, through ligand–receptor signaling, or entering the recipient cells via phagocytosis.

**Figure 5 ijms-20-02853-f005:**
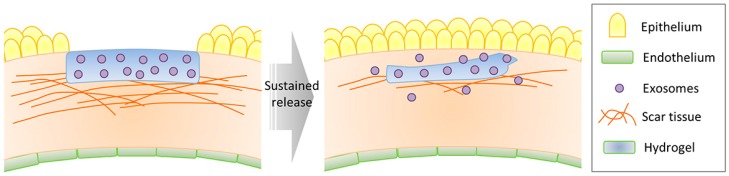
Illustration of sustained delivery of exosomes for the treatment of corneal scarring and neovascularization. The biodegradable hydrogel, placed on the bare stroma, provides first-line protection for the encapsulated exosomes from proteolytic degradation and allows sustained release of the exosomes as it gradually degrades over a period of time.

**Table 1 ijms-20-02853-t001:** An overview of studies reporting MSC differentiation to corneal cell types. **Abbreviations:** MSC, mesenchymal stem cells; CK3, cytokeratin3; CK12, cytokeratin12; Kera, keratocan; Lum, lumican; MET, mesenchymal-epithelial transition; GSK3, glycogen synthase kinase 3; TGFβ, transforming growth factor beta; KS, keratan sulfate; ALDH, aldehyde dehydrogenase; TSG-6, tumor necrosis factor-α stimulated gene/protein 6.

Corneal Tissues	MSC Source	In Vitro Study and Outcomes	In Vivo Study and Outcomes	References
**Corneal epithelium**	Rabbit bone marrow	Co-culture with rabbit limbal stem cells, or in conditioned media induced CK3 expression.	Transplantation in fibrin gel to rabbit corneal epithelial defect caused by alkali injury reformed corneal epithelium with CK3 expression.	[50]
	Rat bone marrow	Co-culture with rat corneal stromal cells resulted in CK12 expression.	Transplantation on amnion to rat corneal epithelial deficiency model after alkali injury showed differentiation to epithelial-like cells with CK12 expression.	[38]
	Human bone marrow	-	Transplantation on amnion to rat epithelial defect model after alkali injury inhibited corneal inflammation and angiogenesis; however, CK3 was not detected.	[52]
	Human adipose	Culture in corneal epithelium conditioned media induced CK3 and CK12 expression.	-	[51]
	Human adipose	MET by GSK3 and TGFβ inhibition downregulated mesenchymal genes and up-regulated epithelial genes (E-cadherin, cytokeratins and occludin).	Transplantation of MET cells on fibrin gel to rat total limbal stem cell deficiency model showed expression of human CK3, 12 and E-cadherin on rat corneal surface (unpublished).	[55]
	Mouse bone marrow	-	Intrastromal injection to Kera knockout murine model expressed human Kera and cells assumed keratocyte phenotype without immune or inflammatory response.	[62]
**Corneal stroma**	Rabbit adipose	-	Application on a polylactic-co-glycolic acid scaffold to mechanically induced rabbit stromal defect induced differentiation to Kera and ALDH3A1 expressing cells.	[68]
	Human bone marrow	Culture in keratocyte conditioned medium to express keratocyte markers (ALDH1A1, Lum and Kera).	-	[63]
	Human adipose	Culture in reduced serum condition with insulin and ascorbate induced the expression of stromal matrix components (Kera, KS, ALDH3A1).	-	[66]
	Human dental pulp	Culture in keratocyte differentiation medium induced Kera and KS proteoglycan expression.	Intrastromal injection to mouse corneal stroma did not affect corneal transparency and absence of immune rejection, with production of stromal ECM components (human type-1 collagen and Kera).	[79]
	Human periodontal ligament	Induction by growth factors in stromal environment generated CSK-like cells. Intrastromal injection to porcine corneas followed by organ culture showed human CD34, ALDH3A1, Kera and Lum expression.	-	[83]
	Human corneal stromal stem cells	Pellet culture in serum-free condition induced Kera, KS and ALDH3A1 expression. Induction culture in keratocyte differentiation medium resulted in ALDH3A1 and Kera expression.	Intrastromal injection to Lum knockout murine corneas re-expressed human Kera and Lum, and corneal transparency was improved. Transplantation of cells in fibrin gel to murine debridement wound resulted in wound healing and transparent corneas with human Kera expression. Topical application of human cells in fibrinogen to murine corneal epithelium debridement model suppressed inflammation and fibrosis through TSG-6 dependent mechanism.	[65,67,69]
**Corneal endothelium**	Human umbilical cord	Differentiation in medium containing GSK3β inhibitors induced the expression of Na^+^K^+^ATPase.	Cell injection to rabbit bullous keratopathy model improved corneal thickness and transparency with Na^+^K^+^ATPase expression.	[91]

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
