# Peer review of "Current Trends and Future Perspective of Mesenchymal Stem Cells and Exosomes in Corneal Diseases"

_ijms, 2019, doi:10.3390/ijms20122853_

Round 1

Reviewer 1 Report

Authors describe the potential applications of MSC and MSC-Exo in eye diseases.

The review is well wrtitten and designed. References are updated.

Author Response

 Thank you for the positive comments.

Reviewer 2 Report

In a well-written and comprehensive study, Mehta's group reviewed the current studies on MSC for corneal reconstruction. It would significantly add to this review if the authors add/expand the following subjects/concepts.

Problems with reproducibility of any cell therapy including MSCs.

Current potency assays for MSCs.

The differences between MSCs derived from different tissues.

The difficulty of storage and maintenance of MSCs and their exo and what are the current procedures to circumvent this problem.

Contents of exo and potential targets/mechanisms to focus on for the translation of exo to clinical use.

Please add the recent clinical trial that compares MSCs with CLET.

Minor points:

Please add more illustrations and photos.

Ref 10 is not properly cited.

Author Response

Thank you for the positive comments. New paragraphs have been added on different aspects as suggested.

1. Problems with reproducibility of any cell therapy including MSCs.

A new section of “7. Challenges of MSC therapy” has been added on P11-12.

2. Current potency assays for MSCs.

A new section of “8. MSC potency for translational use” has been added on P12-13.

3. The differences between MSCs derived from different tissues.

The new information has been added on P3-P4. A new Fig. 2 describes different sources of MSC.

4. The difficulty of storage and maintenance of MSCs and their exo and what are the current procedures to circumvent this problem.

The new information has been added on P16.

5. Contents of exo and potential targets/mechanisms to focus on for the translation of exo to clinical use.

A new section of “13. The cargo of MSC-Exo and potential mechanisms for therapeutic effects” has been added on P16.

6. Please add the recent clinical trial that compares MSCs with CLET.

This point has been added on P6.

Minor points:

Please add more illustrations and photos.

We have added 3 new figures. They include

Figure 1. Overview of human cornea and its pathological opacification,

Figure 2. Major MSC sources in human tissues, and

Figure 5. Illustration of sustained delivery of exosomes for the treatment of corneal scarring and neovascularization

Ref 10 is not properly cited.

Correction has been made on P3 (revised Ref 12).